# Selective Etching of Si versus Si_1−x_Ge_x_ in Tetramethyl Ammonium Hydroxide Solutions with Surfactant

**DOI:** 10.3390/ma15196918

**Published:** 2022-10-05

**Authors:** Yongjoon Choi, Choonghee Cho, Dongmin Yoon, Joosung Kang, Jihye Kim, So Young Kim, Dong Chan Suh, Dae-Hong Ko

**Affiliations:** 1Department of Materials Science and Engineering, Yonsei University, Seoul 03722, Korea; 2BIO-IT Micro Fab Center, Yonsei University, Seoul 03722, Korea

**Keywords:** silicon, silicon-germanium, gate-all-around device, selective wet etching, epitaxial growth, multi-layer

## Abstract

We investigated the selective etching of Si versus Si_1−x_Ge_x_ with various Ge concentrations (x = 0.13, 0.21, 0.30, 0.44) in tetramethyl ammonium hydroxide (TMAH) solution. Our results show that the Si_1−x_Ge_x_ with a higher Ge concentration was etched slower due to the reduction in the Si(Ge)–OH bond. Owing to the difference in the etching rate, Si was selectively etched in the Si_0.7_Ge_0.3_/Si/Si_0.7_Ge_0.3_ multi-layer. The etching rate of Si depends on the Si surface orientation, as TMAH is an anisotropic etchant. The (111) and (010) facets were formed in TMAH, when Si was laterally etched in the <110> and <100> directions in the multi-layer, respectively. We also investigated the effect of the addition of Triton X-100 in TMAH on the wet etching process. Our results confirmed that the presence of 0.1 vol% Triton reduced the roughness of the etched Si and Si_1−x_Ge_x_ surfaces. Moreover, the addition of Triton to TMAH could change the facet formation from (010) to (011) during Si etching in the <100>-direction. The facet change could reduce the lateral etching rate of Si and consequently reduce selectivity. The decrease in the layer thickness also reduced the lateral Si etching rate in the multi-layer.

## 1. Introduction

The structural development of transistors has been extensively studied to improve device performance. Gate-all-around field effect transistors (GAAFETs), which cover all sides of a channel while being in contact with the gate, are considered the most promising candidate due to their exceptional performance [1,2,3,4]. The hole mobility of pMOS can be increased by replacing Si_1−x_Ge_x_ with a channel material instead of Si [5,6]. To create a structure with a horizontal Si_1−x_Ge_x_ channel, only Si should be selectively removed from the Si_1−x_Ge_x_/Si/Si_1−x_Ge_x_ multi-layer. Selective etching in the multi-layers could be influenced by structural parameters, such as the etched layer thickness and the surface orientation [7,8]. The exposed Si_1−x_Ge_x_ surface should be smooth after Si selective etching, as the surface roughness degrades device performance. Thus, the study on Si-selective etching versus Si_1−x_Ge_x_ has become increasingly important in the fabrication of a Si_1−x_Ge_x_ channel for GAAFET.

Si and Si_1−x_Ge_x_ etching methods have been investigated with various etchants, including acid mixtures of HNO_3_/HF/CH_3_COOH (HNA) [7] or alkaline KOH [9] and tetramethyl ammonium hydroxide (TMAH: (CH_3_)_4_N(OH)) [9]. When using HNA mixtures as etchants, Si_1−x_Ge_x_ is etched faster than Si [7]. Conversely, the Si etching rate is higher than that of Si_1−x_Ge_x_ in KOH and TMAH etchants. However, the handling of KOH can be difficult due to K^+^ ionic contamination [10]. TMAH has thus attracted attention as a promising Si-selective etchant for Si_1−x_Ge_x_ GAAFETs. Although several studies have reported a higher etching rate for Si compared to Si_1−x_Ge_x_ in TMAH [11,12], there has been no investigation of the selectivity mechanism through a chemical analysis of the surface. Furthermore, the effect of etching direction on selectivity in the multi-layers has not been investigated.

Several studies have been conducted on the reaction of Si etching in TMAH [9,12,13,14]. The Si etching in alkaline is caused by hydroxide ions (OH^–^) attaching to the Si atom on the surface and breaking back bonds. Polycrystalline Si will be isotropically etched in TMAH because the average number of bonds at the surface is the same, regardless of orientation. However, crystalline Si is anisotropically etched in TMAH because the number of back bonds that must be broken for etching differs depending on the surface orientation. Si atoms on the (001) surface have two dangling bonds and two back bonds, while Si atoms on the (111) surface have one dangling and three back bonds. Therefore, the (001) surface that requires the breaking of two back bonds is etched faster than the (111) surface that requires the breaking of three back bonds. Etching anisotropy can lead to the unintended undercut forming the facet. This undercut can be controlled by the addition of Triton X-100 (C_14_H_22_O(C_2_H_4_O)_n_, where n = 9–10), a surfactant with both hydrophobic and hydrophilic parts in its molecules. Specifically, the Si undercutting can be greatly reduced even if only 0.1 vol% Triton is added to 25 wt% TMAH [15,16]. The addition of Triton was also reported to reduce the Si surface roughness after the wet etching of Si in TMAH [15]. However, studies on the effect of Triton on the etch rate and surface roughness of Si_1−x_Ge_x_ has not been reported to date. In addition, the effect of Triton on Si-selective etching in a multi-layer with different pattern directions has not been investigated.

In this study, we investigated Si selective etching versus Si_1−x_Ge_x_ in TMAH with and without Triton X-100. When the Ge concentration of Si_1−x_Ge_x_ increases, the formation of –OH bonds on the surface is insufficient, resulting in a decrease in the etching rate. In the Si/Si_1−x_Ge_x_ multi-layer, Si was selectively removed due to the difference in the etching rate compared to Si_1−x_Ge_x_. We also investigated the structural effects, such as the surface orientation and the thickness of the etched layer, on Si selective etching. TMAH has been proven to be an anisotropic etchant, with different etching rates depending on the surface orientation [14]. When Si was etched in the multi-layer, the facet formation differed according to the lateral etching direction. Moreover, the addition of Triton to TMAH could affect the anisotropy and facet formation in Si-selective etching.

## 2. Materials and Methods

Figure 1 shows the etching process of single-crystalline Si and Si_1−x_Ge_x_ in single- and multi-layer structures grown on the (001) wafer (Wafer Biz, Seoul, Korea). The directions of Si etching in TMAH are indicated by the red arrows in Figure 1. The oxide layer grown on the (001) wafer was patterned in the form of alternate 5 µm lines and spaces in the <110> and <100> directions (Figure 1a). The Si substrate was partially exposed as buffered oxide etchant removed the 165-nm-thick SiO_2_ layer. The Si substrate was partially exposed as buffered oxide etchant removed the oxide. Si can be selectively removed because SiO_2_ is etched almost four orders of magnitude slower than Si (001) in TMAH [13]. The surface orientation of the facet formed under the SiO_2_ layer during the etching process was obtained through the angle between the (001) plane. The facets that can be formed when Si is etched are planes that exist on the zone axis in the pattern direction.

Si_1−x_Ge_x_ layers with four different Ge concentrations of less than 50% were prepared in order to investigate the effects of Ge concentration on the etching rate of the Si_1−x_Ge_x_ layers. When the Ge concentration of Si_1−x_Ge_x_ is more than 50%, the lattice mismatch between Si and Si_1−x_Ge_x_ is too large to cause defects. To exclude the defects that may affect etching, Si_1−x_Ge_x_ with a Ge concentration of less than 50% was prepared. The Si and Si_1−x_Ge_x_ layers were epitaxially grown on Si (100) wafers using ultra-high-vacuum chemical vapor deposition (UHV-CVD, Jusung, Engineering Co., Ltd., Gwangju, Kyeonggi, Korea). The native oxides of the surface were removed using a 1% HF solution before the deposition of the epitaxial layers, and Si_2_H_6_ and GeH_4_ were utilized as the precursors for Si and Ge, respectively. The Ge concentration of Si_1−x_Ge_x_ single-layer used for etching was measured by high-resolution X-ray diffraction (HR-XRD). Figure 2 shows the (004) rocking curves of the Si_1−x_Ge_x_ single-layers with four different Ge concentrations. The Ge concentrations of the Si_1−x_Ge_x_ layers calculated from the peak positions were 0.13, 0.21, 0.30, and 0.44. We also found that the rocking curves show the well-defined fringes, which mean the abrupt interface. The Ge concentrations measured by the spectroscopy ellipsometer were similar to the results of XRD analysis. The epitaxial Si_1−x_Ge_x_ single layers for vertical etching were grown to thicknesses of 40, 37, 25, and 22 nm for Si_0.87_Ge_0.13_, Si_0.79_Ge_0.21_, Si_0.70_Ge_0.30_, and Si_0.56_Ge_0.44_, respectively (Figure 1b).

For lateral etching, Si/Si_1−x_Ge_x_ multi-layers with different thicknesses of Si layers were deposited to investigate the structural effects (Figure 1c). Si layers were deposited with thicknesses ranging from 10 to 100 nm between the Si_1−x_Ge_x_ layers to form Si_1−x_Ge_x_/Si/Si_1−x_Ge_x_ multi-layer structures. Figure 3 shows transmission electron microscopy (TEM) image of the Si/Si_1−x_Ge_x_ multi-layer and the distribution of Si and Ge elements obtained by energy dispersive spectroscopy (EDS) mapping. In Figure 3a, we confirmed that the interfaces between the layers were sharp and there were no defects. Figure 3b,c shows that there is no Ge atom in the Si layer, indicating that the gas precursor was controlled for each layer in the deposition of the Si/Si_1−x_Ge_x_ multi-layer. Each element of interest has distinct X-ray emission spectra, as shown in Figure 3d. Si_1−x_Ge_x_ layers in the multi-layer were grown using the same recipe as the Si_0.7_Ge_0.3_ single layer. Figure 4 shows the results of the line detection of Si and Ge profiles in the multi-layer using TEM EDS analysis. Ge concentration was measured at about 29.6%, which is the same as the XRD result of the Si_0.7_Ge_0.3_ single layer. The multi-layer structures were coated with a photoresistant agent and exposed to ultraviolet radiation using a pattern with alternating 5 µm lines and spaces. To expose the sidewalls of the multi-layer, reactive ion etching (RIE) was used for patterning. The sidewalls revealed by RIE in the <110>- and <100>-directional patterns were the (110) and (010) planes, respectively. The surface orientation of the facet formed during the lateral etching of Si was obtained through the angle between the interface of the Si_0.7_Ge_0.3_ layer.

TMAH (Sigma-Aldrich, Darmstadt, Germany) and Triton X-100 (Sigma-Aldrich, Darmstadt, Germany) were used as the etchant and additive, respectively. TMAH with and without Triton was employed to investigate the effects of Triton on the etching anisotropy and selective etching of Si. The etchants were stirred at 400 rpm for 1 h until they reached 60 °C. The etching process was then carried out at 60 ± 1.0 °C with stirring at 100 rpm. After that, all specimens were immediately rinsed twice using deionized water.

The Ge concentrations of the Si_1−x_Ge_x_ layers were obtained from the (004) rocking curves measured by HR-XRD using a Rigaku SmartLab (Rigaku, Tokyo, Japan) with Cu Kα1 (λ = 1.5406 Å) with a Ge (220) × 2 monochromator (Rigaku, Tokyo, Japan). The elemental composition of Si/Si_1−x_Ge_x_ multi-layer was explored using TEM (JEM-2100F, JEOL Ltd., Tokyo, Japan) EDS analysis. The thicknesses of the Si_1−x_Ge_x_ single layer and Si/Si_0.7_Ge_0.3_ multi-layers were measured using a spectroscopic ellipsometer (J.A. Woollam Co., Lincoln, NE, USA) and TEM. The chemical composition of the etched Si and Si_1−x_Ge_x_ surfaces was identified by normal angle X-ray photoelectron spectroscopy (XPS) with the Al Kα µ-focused monochromator (Thermo Fisher Scientific Inc., Waltham, MA, USA). The surface roughness after the etching process was evaluated using atomic force microscopy (AFM, XE-100, Park Systems Corp., Suwon, Kyeonggi, Korea). The dimensions of the laterally etched Si layers were measured with SEM (JSM-7001F, JEOL Ltd., Tokyo, Japan) and TEM.

## 3. Results and Discussion

We etched the line-patterned oxide (001) wafers in 25 wt% TMAH with and without 0.1 vol% Triton at 60 °C to investigate the effects of surface orientation and additive on the etch rate of Si. The 165 nm SiO_2_ layer on the Si (001) wafer was partially removed to reveal the Si substrate at 5 μm intervals in the <110> and <100> directions. Si can be selectively removed because SiO_2_ is etched almost four orders of magnitude slower than Si (001) in TMAH [13].

Figure 3 shows cross-sectional images when Si was etched by TMAH on the oxide wafers patterned in <110> and <100> directions. Figure 5a,b shows the microstructure of the <110>-patterned oxide wafer etched by TMAH etchant with and without Triton. As shown in Figure 5a, the (111) facet, which is the plane with the slowest etch rate on the <110> zone axis, is formed at an angle of about 55° with respect to the (001) substrate [13]. This result is consistent with the findings of the previous study [9,13,14], which revealed that the etching rate of the (111) plane is the slowest in alkaline etchant. As shown in Figure 5b, the (111) facet was maintained even when Triton was added to TMAH. Figure 5c,d depicts the microstructure of the <100>-patterned oxide wafer etched by a TMAH etchant with and without 0.1 vol% Triton. As shown in Figure 5c, Si was etched in the (001) substrate direction and in the direction perpendicular to that direction. On the <100>-zone axis, the plane perpendicular to the (001) plane was the (010) plane. The formation of this facet can be altered by the addition of Triton. As shown in Figure 5d, a facet formed after etching with a 45° angle between the substrates was the (011) surface on the <100>-zone axis.

Table 1 shows the Si etching rates in each direction using TMAH with and without Triton. In the <110>-patterned oxide wafer, the (001) surface was etched at a rate of 133.7 nm min^−1^, whereas the (111) surface was etched at a relatively slow rate of 16.2 nm min^−1^, forming a (111) facet. When Triton was added in TMAH, the formed facet plane was maintained, but the Si etching rates of the (001) and (111) surfaces were reduced to 123.3 nm min^−1^ and 14.0 nm min^−1^, respectively. The decrease in the etch rate is because the hydrophobic part of the Triton molecule attached to the Si surface, preventing the reactants from reacting with the Si atoms. This behavior aligns with the previous work reporting on the role that the surfactant plays as a filter in controlling the surface reactivity [17]. In the <100>-patterned oxide wafer, the etching rates of (001) and (010) surfaces in TMAH were similar, at 136.1 nm min^−1^ and 140.3 nm min^−1^, respectively. This result indicates that the etching rate of (001) planes was the slowest in the <100>-zone axis in the TMAH etchant. As shown in Figure 5d, however, the (010) facet changed to the (011) facet because the (011) plane became the plane with the slowest etching rate at 58.8 nm min^−1^ in the <100>-zone axis by Triton. For this behavior, Gosálvez et al. reported that Si (110) is very reactive and the adsorption density is very high, whereas Si (100) is less reactive and the adsorption density is relatively low [17]. Therefore, when adding 0.1 vol% Triton to 25 wt% TMAH, the etching rate of Si (110) became lower than that of Si (001).

To selectively remove Si from the Si_1−x_Ge_x_/Si/Si_1−x_Ge_x_ multi-layer structure, we studied the etching of Si and Si_1−x_Ge_x_. The etching rates of Si and Si_1−x_Ge_x_ were compared at various Ge concentrations. Figure 6 shows the vertical etching rates of Si and Si_1−x_Ge_x_ (x = 0.13, 0.21, 0.30, and 0.44) in TMAH, with and without Triton at 60 °C. The vertical etch rate was determined by the slope of the etched amount that linearly increases over etching time. The etched thickness of Si_1−x_Ge_x_ layer was obtained by measuring the difference in the thickness before and after etching with a spectroscopic ellipsometer. The vertical etching rates of Si_1−x_Ge_x_ decreased with increasing Ge concentration, whether or not Triton was added. This result is consistent with the previous study, showing that the etching of Si_0.7_Ge_0.3_ is slower than that of Si in NH_4_OH, TMAH, and tetraethyl ammonium hydroxide alkaline etchants [18]. The reduction in the Si etching rate achieved by adding Triton to TMAH was less than 20%. However, as the Ge concentration increases, the rate reduction increases by more than 20%. In the case of Si etching, when Triton molecules were adsorbed on the surface, the vertical etching rate was reduced because Triton prevented OH^–^ from reaching the surface Si atom. The low etching rate of Si_1−x_Ge_x_ with high Ge concentration indicates that there is a change in the etching reaction. However, the mechanism of selective etching has not been fully understood.

Therefore, we analyzed the etched surface using XPS to better understand the mechanism of Si and Si_1−x_Ge_x_ etching. All Si and Si_1−x_Ge_x_ specimens were cleaned in HF to remove native oxides before etching. The cleaned specimens were then etched in 25 wt% TMAH for 2 min. Figure 7 presents the relative atomic percentages of Si 2p, Ge 2p, O 1s, and C 1s on the surface before and after etching in TMAH. About 5% of the C 1s detected in all samples is attributed to air exposure contamination prior to loading into the XPS equipment. This also shows that C atoms of TMAH were well removed by rinsing after etching. The O 1s detected on the HF-cleaned surface originated from the native oxide formed before XPS loading. After etching in TMAH, the percentages of O 1s on the Si and Si_1−x_Ge_x_ surfaces increased, indicating the occurrence of surface oxidation. The oxidation is expected to result from the reaction between OH^–^ ions of alkaline TMAH and the Si(Ge) atoms on the surface. The increase in the O 1s percentage was the largest in Si and roughly decreased with the increasing Ge concentration. The difference in the O 1s percentage increase with Ge concentration is correlated with the etching selectivity.

To understand the oxidation step, we further investigated the bonding between Si(Ge) and O through the O 1s core-level spectra of Si and Si_1−x_Ge_x_ after etching in TMAH. Figure 8 shows the O 1s XPS spectra of surfaces of as-grown and etched Si_1−x_Ge_x_ with Ge concentrations of 0, 0.13, 0.21, 0.30, and 0.44, respectively. For all the specimens, the peak can be deconvoluted into three components. In the literature, it has been proposed that the O 1s peak on the Si surface consists of at least four components, such as Si–O–Si, Si–OH, non-bridging O (Si–O–Na, K), and chemisorbed O [19,20]. Carniato et al. reported that the Si–OH peak at 532.4 eV and the Si–O–Si peak at 531.4 eV were located in the O 1s spectra of the water-reacted Si surface [21]. For the electrons around the O atom, the density is higher in Si–O–Si than in Si–OH because H is more electronegative than Si. These electrons cause an increase in Coulomb repulsion with each other, making the binding energy of Si–O–Si lower than that of Si–OH. In Figure 8, there are two high peaks (Si–O–Si and Si–OH) and a small peak in chemisorbed O, but the non-bridging O peak does not exist due to the absence of ions, such as Na^+^ and K^+^ in TMAH solutions.

Figure 8b shows the O 1 s XPS spectra of etched surfaces of Si_1−x_Ge_x_ with Ge concentrations of 0, 0.13, 0.21, 0.30, and 0.44, respectively. The spectra show that O 1s peak intensity increased compared to as-grown Si_1−x_Ge_x_. It also shows that the O 1s peak position shifted to the left with decreasing Ge concentration. The peak increment and shift can be explained by comparing the deconvoluted O 1s spectra of Si and Si_1−x_Ge_x_ etched in TMAH. As Ge concentration decreased, the intensity of the Si(Ge)–O–Si(Ge) peak decreased, while that of the Si(Ge)–OH peak, which has a larger binding energy, increased; thus, the O 1s peak shifts to the left. In the Si etching process, hydroxyl bond (–OH) is formed as electrons are injected into the conduction band of Si from OH^−^ to oxidize Si [22]. Then, the Si–Si bond opposite the Si–OH bond was weakened due to the presence of O, which is more electronegative than Si. The weakened Si–Si bond could easily be replaced by another OH^−^ and etched into water-soluble Si(OH)_4_. Therefore, the formation of Si–OH bonds by oxidation is necessary for Si to be etched. Although the mechanism of Si_1−x_Ge_x_ etching in alkaline-based solutions has not been well studied, the process involves oxidation by OH^−^, as described in the case of Si etching. In the literature, when Si_1−x_Ge_x_ is oxidized during initial etching, Ge is pulled out and this effect is considerably larger in the surface region [23]. This phenomenon can explain why the composition of Ge 1s decreased at a larger rate than that of Si 2p after etching, as shown in Figure 7. The increase in the intensity of the Si(Ge)–O–Si(Ge) peak in Si_1−x_Ge_x_ suggests that O fills the place where Ge is pulled out, forming the bridging O (Si–O–Si). As mentioned above, Si and SiO_2_ have a very high etch selectivity and, thus, the formation of Si–O–Si prevents etching. Therefore, the difference in etching rate between Si and Si_1−x_Ge_x_ is determined by the ratio of Si(Ge)–O–Si(Ge) and Si(Ge)–OH intensities after etching.

We also investigated the morphology of the etched surface. Table 2 summarizes the root mean square (RMS) values obtained by AFM for the roughness of the Si and Si_1−x_Ge_x_ surfaces etched in TMAH. The RMS values of the as-grown layers were less than 0.2 nm. Table 2 shows that the etched surface of Si is rougher than that of Si_1−x_Ge_x_. As the Ge concentration increases, the etched surface of Si_1−x_Ge_x_ becomes smoother. The effect of Ge concentration on the roughness of the etched surface can be explained by the XPS analysis shown in Figure 7 and Figure 8. The bridging Si(Ge)–O–Si(Ge) on the surface of Si_1−x_Ge_x_ with high Ge concentration interfered with the formation of Si(Ge)–OH, so that the Si_1−x_Ge_x_ was uniformly etched. Table 2 also shows that the addition of Triton to TMAH reduced the surface roughness of Si and Si_1−x_Ge_x_. These results are consistent with the literature that revealed that the roughness of the Si surface can be reduced by the addition of Triton to TMAH [15]. The Triton layer covering the surface prevents H_2_ production and breaks the H_2_ bubble before it becomes too large. As a result, the surface morphology is smoothed by reducing the contact inhomogeneities between the surface and the etchant.

To investigate the selective etching of Si, Si_0.7_Ge_0.3_/Si/Si_0.7_Ge_0.3_ multi-layer structures were grown by UHV-CVD. The grown structures were patterned, and the side walls were exposed by RIE. Figure 9 shows the TEM image of the <110>-directionally patterned Si_0.7_Ge_0.3_/Si/Si_0.7_Ge_0.3_ multi-layer structure when Si is selectively removed after 6 m etching in TMAH at 60 °C. Si_0.7_Ge_0.3_ was hardly etched while Si was selectively removed. The Si lateral etching rate is about 20 nm/min, which is much slower than the Si vertical etching rate, which is over 130 nm/min. In addition, Si is etched by exposing the plane at an angle of 55° between the Si_0.7_Ge_0.3_ layers. This result suggests that Si etching in multi-layer is also anisotropic, with different etching rates depending on the surface orientation. Although the side wall of <110>-patterned multi-layer is the (110) plane, the facet is formed on the (111) plane, which has the slowest etching rate of the planes on the <110> zone axis. The facet formation changes the etching rate, so it is important to investigate the structural effect, such as the layer thickness and the etching direction, on the lateral Si etch.

Figure 10 shows the selective etching of Si in the <110>- and <100>-patterned Si_0.7_Ge_0.3_/Si/Si_0.7_Ge_0.3_ multi-layer. The multi-layer structures after etching are different for each pattern in the <110>- and <100>-directions. In the <110>-patterned multi-layer structure, Si was laterally etched, forming (111) facets at an angle of approximately 55° between the Si_0.7_Ge_0.3_ layers (Figure 10a). The (111) planes had the lowest etching rate of the planes on the <110>-zone axis. These results are the same as those shown in Figure 3a and Figure 7. The lateral etched depth depends on the height of the Si layer because of this (111) facet formation. However, in the <100>-patterned multi-layer, Si was perpendicularly etched, as shown in Figure 10b. According to the results in Figure 3c, the (010) planes had the lowest etching rate among the planes on the <100>-zone axis in TMAH. These results indicate that the facet is formed with the slowest etched plane among the planes on a specific zone axis, regardless of the exposed plane before etching. Table 3 shows the etching selectivity between Si and Si_0.7_Ge_0.3_ obtained by Figure 10. The Si etch rate is calculated as the average of the most deeply etched and less etched regions. The Si_0.7_Ge_0.3_ etch rate is evaluated by the amount of thinning of the Si_0.7_Ge_0.3_ layer after etching. In the <110>-patterned multi-layer, the selectivity is relatively low because the facet is formed with (111). In the <100>-patterned multi-layer, however, the etching selectivity is more than 100:1 because the facet is formed on the (010) plane.

Figure 10 and Table 3 also indicate that the Si etching rate decreases as the thickness of the Si layer decreases. The lateral etching rate of the 10 nm thick Si layer is much slower than that of the other layers. To investigate the effects of the sacrificial layer thickness on the selective etching, Si_0.7_Ge_0.3_/Si multi-layers with Si layers of various thicknesses were patterned in the <100> and <100> directions. Figure 11 shows the Si etching rate for each direction in the patterned Si_0.7_Ge_0.3_/Si multi-layers and patterned oxide wafers. The etching rate increases as the thickness of the etched Si layer increases. This behavior is the result of disturbances in the movement of the etchant in the narrow-etched tunnel [7]. When the thickness of the Si layer is over 70 nm, the disturbance in the movement of the etchant is reduced; thus, the etching rate is saturated. In the multi-layer, the lateral etching rate of Si was faster in the <100> direction than in the <110> direction, regardless of the addition of Triton. This difference is because of the (111) facet, which took the longest time to etch in the <110> direction. After adding Triton, the vertical Si(001) etch rate decreased by about 10%. However, the decrease in the lateral etching rate in the <100> direction decreased by 40%–60%, which was more than that of the vertical etching rate. The results in Figure 5d explain the differences in this reduction. The plane with the slowest etch rate changes from (010) to (011) through the addition of Triton. Therefore, the lateral etching rate in the <100>-direction was similar to that of the (011) plane in TMAH with Triton.

## 4. Conclusions

We investigated the selective wet etching of Si and Si_1−x_Ge_x_ in TMAH with and without Triton X-100 in single- and multi-layer structures. During the vertical etching, Si_1−x_Ge_x_ with a higher Ge concentration experiences a lower etching rate because Si(Ge)–OH bonds are hardly formed on the surface of Si_1−x_Ge_x_ due to formation of Si–O–Si. Since the etching rate of Si is higher than that of Si_1−x_Ge_x_, only Si is removed from the multi-layer. In both vertical and lateral etching cases, TMAH is an anisotropic etchant with different etching rates depending on the surface orientation. The addition of Triton not only modulates the surface reactivity but also changes the etching anisotropy. The presence of Triton also changes the facet formation from (010) to (011) during lateral etching in the <100> direction. Therefore, although the addition of Triton reduces the surface roughness, it has the disadvantage of affecting etching selectivity. We believe that the results of the Ge effect on the Si and Si_1−x_Ge_x_ surfaces etched in TMAH will be of great help in experimental studies of Si-selective etching, which plays an important role in technically relevant device fabrication. Furthermore, the results of the Triton effect on etching have made it possible to see, for the first time, Triton’s potential in the formation of 3D structural devices, such as GAAFETs. However, in the study of GAAFET, the investigation on the electrical properties of the Si_1−x_Ge_x_ nano channel has not been fully conducted. Therefore, further work is needed to clarify the electrical properties of Si_1−x_Ge_x_ channels obtained by selectively etching Si in TMAH etchant.

## Figures and Tables

**Figure 1 materials-15-06918-f001:**
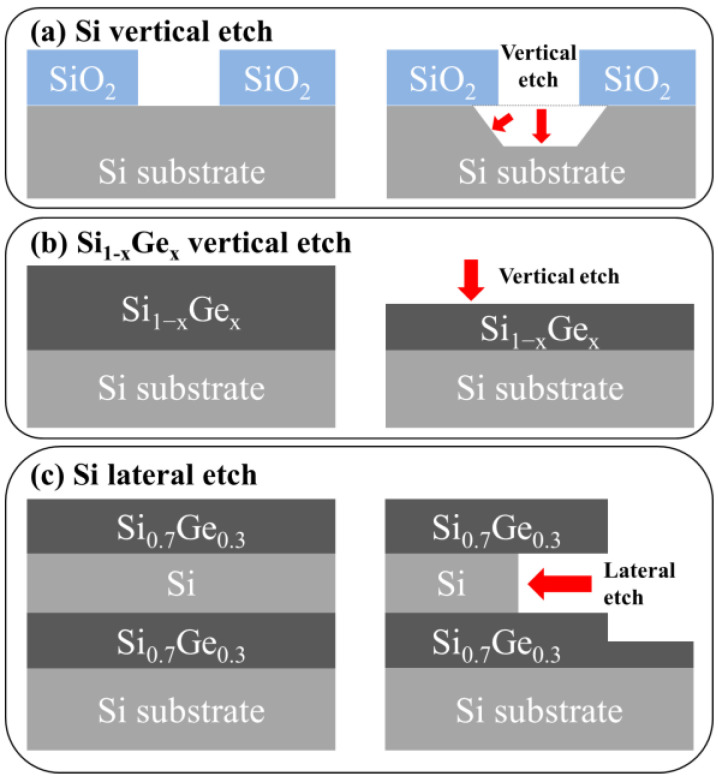
Schematic representation of vertical etching of (**a**) Si, (**b**) Si_1−x_Ge_x_, and (**c**) lateral etching of Si in TMAH etchant.

**Figure 2 materials-15-06918-f002:**
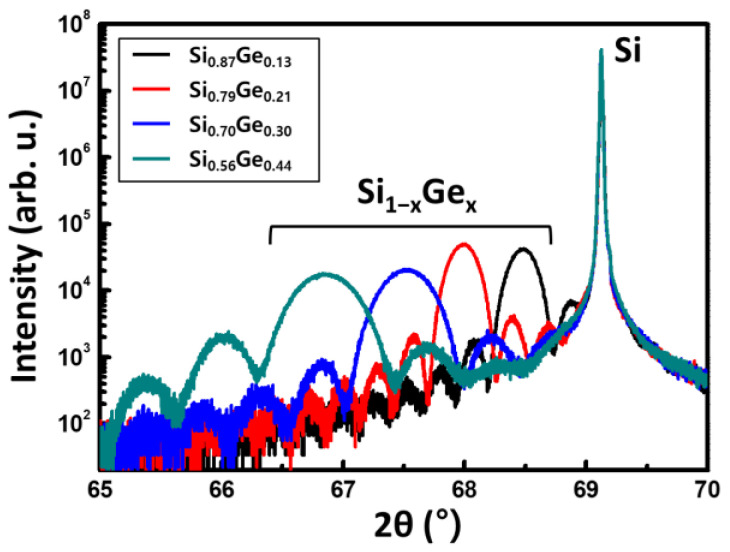
XRD (004) rocking curves of the Si_1−x_Ge_x_ single-layers (x = 0.13, 0.21, 0.30, 0.44).

**Figure 3 materials-15-06918-f003:**
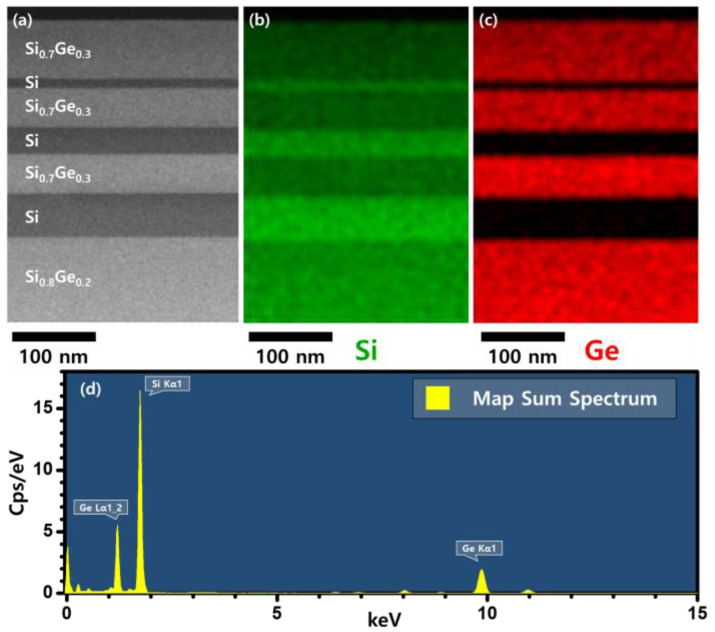
As-grown Si/Si_0.7_Ge_0.3_ multi-layer: (**a**) TEM image, EDS mapping results of (**b**) Si and (**c**) Ge elements, (**d**) X-ray emission spectrum.

**Figure 4 materials-15-06918-f004:**
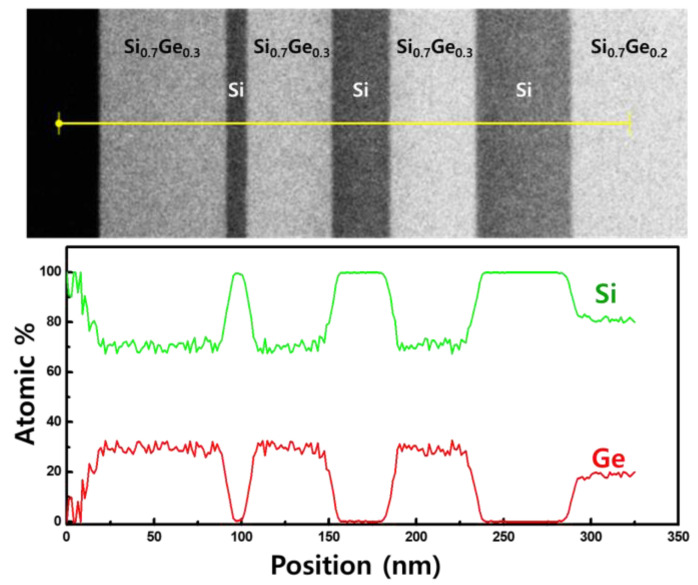
TEM cross-sectional images of Si/Si_0.7_Ge_0.3_ multi-layer and profiles of Si and Ge with EDS line analysis.

**Figure 5 materials-15-06918-f005:**
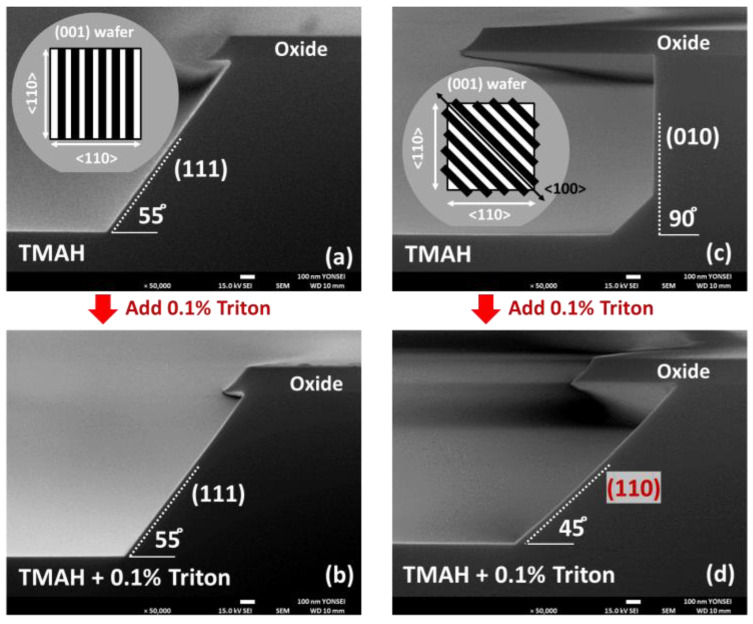
Cross-sectional SEM images of <110>-patterned oxide wafer after 9 min etching in (**a**) TMAH and (**b**) TMAH with 0.1 vol% Triton and <100>-patterned oxide wafer after 9 min etching in (**c**) TMAH and (**d**) TMAH with 0.1 vol% Triton.

**Figure 6 materials-15-06918-f006:**
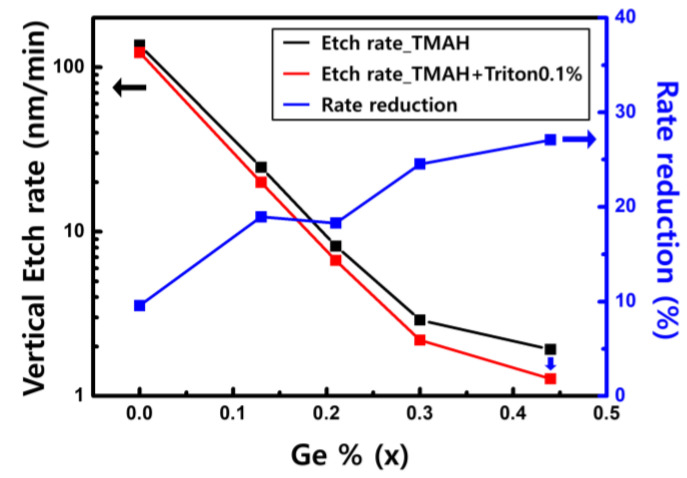
Vertical etch rates of Si and Si_1−x_Ge_x_ in 25 wt% TMAH with and without 0.1 vol% Triton.

**Figure 7 materials-15-06918-f007:**
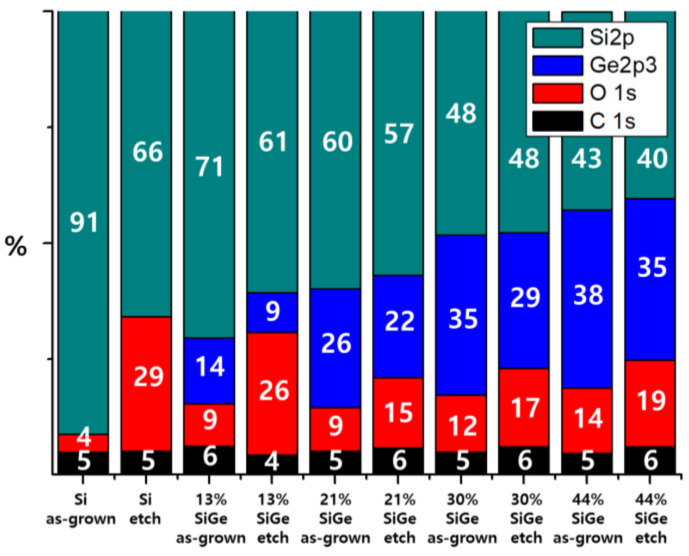
X-ray photoelectron spectroscopy (XPS) surface composition of Si and Si_1−x_Ge_x_ (x = 0.13, 0.21, 0.30, and 0.44) after HF cleaning and TMAH etching.

**Figure 8 materials-15-06918-f008:**
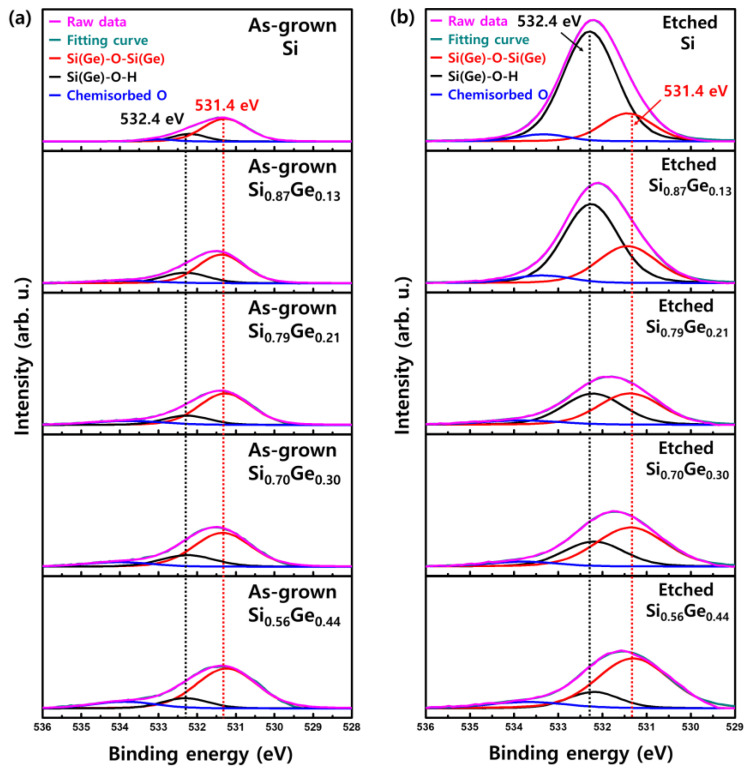
Deconvoluted X-ray photoelectron spectroscopy O 1s spectra of (**a**) as-grown and (**b**) etched Si, Si_0.87_Ge_0.13_, Si_0.79_Ge_0.21_, Si_0.70_Ge_0.30_, and Si_0.56_Ge_0.44_ surfaces.

**Figure 9 materials-15-06918-f009:**
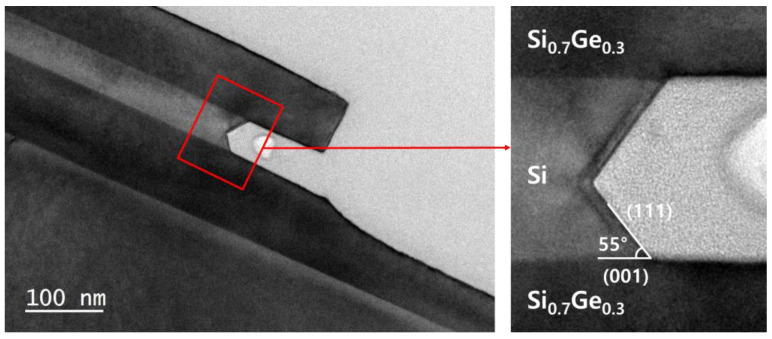
Cross-sectional transmission electron microscopy image of Si_0.7_Ge_0.3_/Si/Si_0.7_Ge_0.3_ multi-layer after selective etching of Si in TMAH.

**Figure 10 materials-15-06918-f010:**
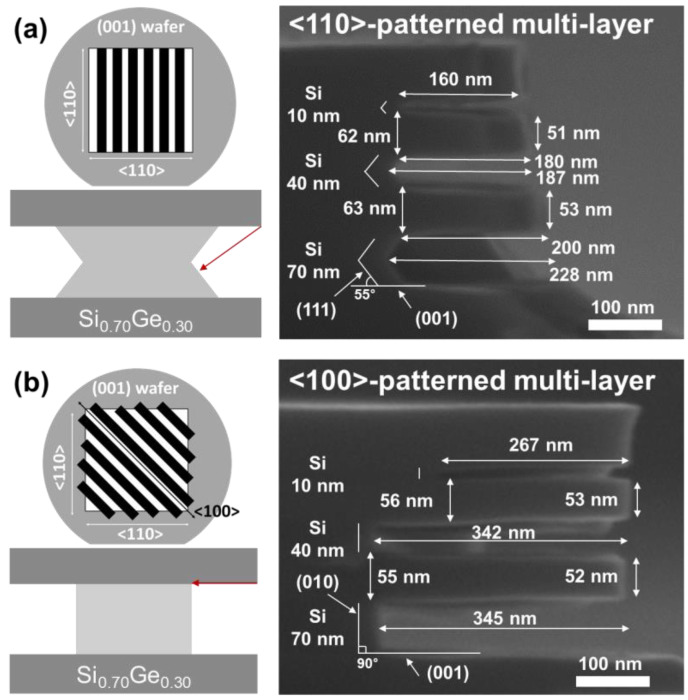
Selective etching of Si layer with various thicknesses in (**a**) <110>-patterned and (**b**) <100>-patterned Si_0.7_Ge_0.3_/Si/Si_0.7_Ge_0.3_ multi-layer using TMAH.

**Figure 11 materials-15-06918-f011:**
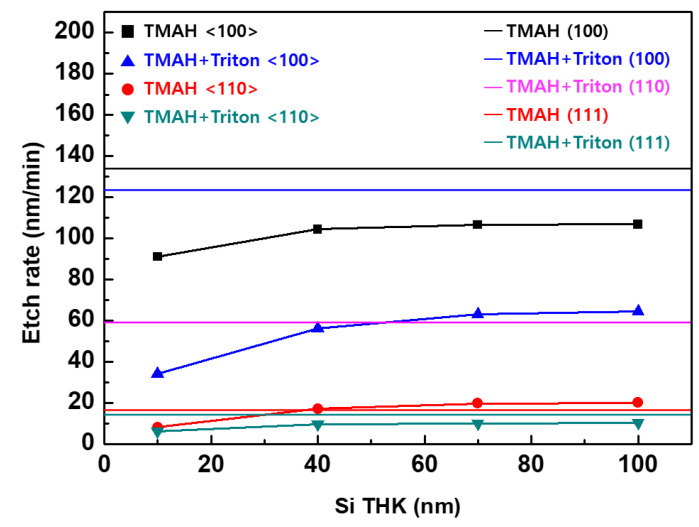
Si etching rates in the multi-layered wafers (symbols) and in the oxide wafers (lines).

**Table 1 materials-15-06918-t001:** Anisotropic etching rates of Si for various surface orientations in pure 25 wt% tetramethyl ammonium hydroxide (TMAH) with and without 0.1 vol% Triton in the patterned oxide wafers.

Surface Orientation	<110>-Pattern		<100>-Pattern	
	TMAH	TMAH + Triton	TMAH	TMAH + Triton
(100)	133.7	123.3	136.1	123.0
(010)	-	-	140.3	-
(111)	16.2	14.0	-	-
(110)	-	-	-	58.8

**Table 2 materials-15-06918-t002:** Root mean square (RMS) values for surface roughness of Si_0.87_Ge_0.13_, Si_0.79_Ge_0.21_, Si_0.70_Ge_0.30_, and Si_0.56_Ge_0.44_ etched in 25 wt% TMAH with and without 0.1 vol% Triton.

Etchant	RMS (nm)			
	Si_0.87_Ge_0.13_	Si_0.79_Ge_0.21_	Si_0.70_Ge_0.30_	Si_0.56_Ge_0.44_
TMAH	1.08	0.51	0.28	0.16
TMAH + Triton	0.70	0.36	0.27	0.12

**Table 3 materials-15-06918-t003:** Etching selectivity of Si:Si_0.7_Ge_0.3_ in the <110>- and <100>-patterned multi-layered wafer.

Etched Layer	Etching Selectivity (Si:S_i0.7_Ge_0.3_)	
	<110>-Pattern	<100>-Pattern
Si 10 nm	30:1	178:1
Si 40 nm	35:1	228:1
Si 70 nm	41:1	230:1

## Data Availability

The data presented in this study are available on request from the corresponding author.

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
