# Peer review of "Selective Etching of Si versus Si1−xGex in Tetramethyl Ammonium Hydroxide Solutions with Surfactant"

_materials, 2022, doi:10.3390/ma15196918_

Round 1

Reviewer 1 Report

Manuscript entitled "Selective Etching of Si versus Si1-xGex in Tetramethyl Ammonium Hydroxide Solutions with Surfactant" by Yongjoon Choi, Choonghee Cho, Dongmin Yoon, Joosung Kang, Jihye Kim, So Young Kim, Dong Chan Suh, and Dae-Hong Ko concerns problems of technological aspect. The research topic is still important in technology.

The authors carried out an investigation of the effect of tetramethyl ammonium hydroxide solutions on Si1-xGex crystals (where x = 0.00, 0.13, 0.21, 0.30, 0.44). The manuscript is well-organized and authors provide comprehensive results obtained by various measurement methods. However, some points need to be clarified/improved before publication:

1.     Highlight the novelty of the report. There is a work in the literature that has already touched on this topic - Virginie Loup et al 2013 ECS Trans. 58 47 (missed in the references of the manuscript).

2.     Enter etching temperature into the text.

3.     103 line: “…the Al Kα µ-focused monochromator 103 using K-alpha…” sounds strange.

4.     The vertical etch rate error should be provided (Fig. 3 and 9).

5.     It would be better for the reader if the figures 5 and 6 were next to each other on the same page.

6.     The abbreviation “A.U.” (in Fig. 5 and 6) means atomic unit it should be "arb. u."

Reviewer 2 Report

In the review of research article titled: Selective Etching of Si versus Si1-xGex in Tetramethyl Ammonium Hydroxide Solutions with Surfactant, authors have presented the research work very well and phenomenon is explained in well scientific way. I would like to see this article publish but some of the minor modifications are necessary before publication, which are as follow,

1-      Why authors have selected the specific window (x = 0.13, 0.21, 0.30, 0.44) for Ge concentration in Si1-xGex, is before this concentration or after is reported somewhere else?

2-      Why in the mentioned window there is no sequence? Are they predicted values of concentrations?

3-      Problem statement of performing this research work is missing in the introduction portion. Please add some drawbacks from previous study which compelled you to perform this work?

4-      Authors are requested to provide the XRD analysis of the study.

5-      Colored elemental mapping will provide the significance of the Ge concentration in the parent material.

6-      It is requested to provide the EDS analysis of all the mentioned concentations to confirm the Ge content in the material. Otherwise this study will lose the readership.

7-      One big question: authors have fabricated the material in multilayered form, but they have not measured any properties of the fabricated material. This is not strong aspect.

8-      It is requested to provide some electronic, electrical or some other properties of the material to make the article stronger for publication purpose.

Reviewer 3 Report

The article presents the results of a study of the selective etching of Si compared to Si1-xGex with different concentrations of Ge (x = 0.13, 0.21, 0.30, 0.44) in a tetramethylammonium hydroxide solution. In general, the presented results represent a rather interesting study, which has both fundamental significance and practical novelty. The article corresponds to the subject of the declared journal and can be accepted for publication after the authors answer the questions.

1. Regarding the data presented in Figure 3, the dependence of the etching rate on the content of germanium in the composition. As can be seen from the presented data, the addition of 0.1% triton at high concentrations does not play a significant role, or the authors should change the presentation on the graph to better see the changes. It should also be reflected exactly how the vertical etch rate data were calculated, taking into account the errors that can level the effect of the difference when adding a triton.

2. The authors should clarify the mechanism of etching in the form of a pyramidal cone with different faces, what exactly is the cause of such etching.

3. The proposed selective etching schemes are related to textural directions; the authors should describe in more detail how the texturing of the samples was carried out. And how polycrystalline samples will behave in this case.

4. The conclusion should reflect in more detail the practical significance of the results obtained and their further application possibilities.

Round 2

Reviewer 3 Report

The authors answered all the questions, the article can be accepted for publication.